# OpenReview forum: "Are Human-generated Demonstrations Necessary for In-context Learning?"
_ICLR.cc/2024/Conference — ICLR 2024 poster_

### Official Review · Reviewer_wk7D · 2023-10-31

**Soundness:** 3 good
**Presentation:** 2 fair
**Contribution:** 3 good
**Rating:** 8
**Confidence:** 4

**Summary:**

This paper proposes Self-Contemplation (SEC) as an alternative for human-generated prompts for in-context learning (ICL) in LLMs. This enables LLMs to generate their own demonstrations for ICL, rather than having to rely on hand-generated human demonstrations. The experiments across a diverse range of different tasks show that SEC can be a meaningful alternative to ICL in some settings, and is able to outperform the zero-shot performance despite not requiring any human demonstrations.

**Strengths:**

- Proposing a method to move beyond hand-crafted demonstrations is a more scalable approach to ICL, where we no longer have to hand-craft domain-specific questions per task setting. Showing that this is able to match human-crafted demonstrations is an interesting finding, and the fact that it generally vastly outperforms the zero-shot setting shows that SEC should generally be used even in settings where we do not have access to high-quality human demonstrations.
- The experimental results seem thorough and cover a wide range of standard tasks for evaluating LLMs.

**Weaknesses:**

-  While finding that it is feasible for a model to generate its own examples (rather than having human-crafted demonstrations) is insightful, the applicability of this method seems limited to large model sizes that are less likely to make reasoning errors during the self-generation process. How much would the method performance drop if tested with less capable language models? The paper would be strengthened with more thorough evaluations of when exactly SEC is usable.
- As a baseline, it would be helpful to have direct comparisons with "Automatic chain of thought prompting in large language models. (Zhang et al., 2022)." to place this method in context relative to prior work. While the authors note that Auto-CoT makes use of questions from the training dataset as few-shot examples, it would still be insightful to see what performance gap this additional information leads to.

In general, paper presentation could be further polished:
- Nit: Clean up Figure 1 further – e.g. typos follwing -> following , Demonetration -> Demonstration, in the Input to LLM section for SEC Demonstration Generation
- Nit: Table 2 is a little hard to read, maybe have larger spaces / break lines between different rows of strategies

**Questions:**

The authors note that ICL and SEC get different questions correct for GSM8K. Have they tried combining both methods? In other words, using ICL with human-crafted demonstrations combined with having the model additionally generate its own questions. Does this lead to better coverage in the correctly answered questions?

---

> ### Author Response · Authors · 2023-11-13
>
> We thank the reviewer for their insightful comments and constructive feedback. We address each concern below:
>
>
>
> **Q1:** SEC’s performance on less capable language models.
>
>   **A1:** We appreciate your concern about SEC’s applicability to smaller, less capable models. As reported in Section 4, SEC indeed shows a performance drop when applied to smaller models, which can struggle with instruction following and generating high-quality few-shot examples. This is a critical aspect of our ongoing research, and we plan to extend our evaluations to a wider range of models, including open-source ones. These additional experiments will provide deeper insights into SEC's scalability and effectiveness across different model sizes.
>
> However, it is important to highlight the growing trend towards larger and more capable language models in the field. As language models continue to scale up in size and sophistication, we anticipate that the importance and relevance of methods like SEC will increase correspondingly. The ability of SEC to autonomously generate demonstrations is likely to become more valuable as models become more adept at complex reasoning and generation tasks. Therefore, while we acknowledge the current limitations with smaller models, we believe that the scaling trend in language models will make SEC an increasingly crucial method in the future.
>
> **Q2:** Comparisons with "Automatic chain of thought prompting in large language models (Zhang et al., 2022)"
>
> **A2:** We recognize the importance of contextualizing SEC within the landscape of related work, such as Auto-CoT by Zhang et al. Our initial decision to exclude Auto-CoT from our baseline comparisons stemmed from its intensive querying and clustering requirements to generate demonstrations. This process can be particularly challenging, and even infeasible, for users dealing with single-question scenarios, which are common in practical applications.
>
> However, we appreciate the reviewer's suggestion to include a comparison with Auto-CoT. In our revised manuscript, we plan to incorporate a direct comparison between SEC and Auto-CoT on some classical benchmarks.
>
> **Q3:** Combining ICL and SEC
>
> **A3:** The suggestion to combine ICL with human-crafted demonstrations and SEC’s model-generated questions is indeed intriguing. While we have not explored this combination, we recognize its potential for synergistically enhancing performance. We will initiate experiments combining these methods in the coming days.
>
> However, it's important to note that the primary contribution of this paper remains the introduction of SEC as a strong, novel and flexible zero-shot prompting method.
>
> **Q4:** Presentation and Figures
>
> **A4:** We thank the reviewer for pointing out the typographical errors in Figure 1 and the readability issue in Table 2. We will correct these in our revised version to enhance clarity and presentation quality.
>
> Once again, we thank you for your valuable feedback and look forward to further improving our work.

---

> > ### Comment · Reviewer_wk7D · 2023-11-22
> >
> > I appreciate the author response with additional experiments addressing my concerns. In particular, the baseline with Auto-CoT is helpful and the added experiment on an underrepresented task shows an interesting limitation of the method. I see that the authors are also revising the main contributions of the paper for clarity and focus on framing the method as a zero-shot method. I would also suggest updating the title to make this more clear. Overall, I am happy to update my score given these additional changes being incorporated into the paper.

---

> ### Author Response · Authors · 2023-11-18
> **Comparison between SEC and Auto-CoT**
>
> We have carried out the experiments on Auto-CoT. CoT-SEC's performance is comparable to Auto-CoT, even without access to the full test dataset and additional clustering.
>
> |                        |   GSM8K  | ARC |
> |---------------------- |------|-------|
> |                       |      |       |
> | *Zero-shot CoT*       |   73.4 | 84.1  |
> | *CoT-ICL*             | 77.4 | 87.9  |
> | *Auto-CoT*            |77.5|87.8|
> | *CoT-SEC*             | 77.0|86.9|
> |                      | (-0.5)|(-0.9) |
>
> The results in the table will later be added into the revised version of our paper.

---

### Official Review · Reviewer_B8T2 · 2023-10-31

**Soundness:** 2 fair
**Presentation:** 3 good
**Contribution:** 3 good
**Rating:** 6
**Confidence:** 4

**Summary:**

This paper proposed a novel zero-shot learning framework, self-contemplation prompting strategy (SEC), that uses LLMs to generate demonstrations on a task and apply in-context learning (LCL) with the LLM-crafted demonstrations.
Interestingly, extensive experiments in arithmetic reasoning, commonsense reasoning, multi-task language understanding, and code generation benchmarks, show that the SEC performance is competitive or outperforms ICL with the human-crafted demonstrations.

**Strengths:**

This is very interesting in the sense that it completely eliminates the effort of writing demonstrations in in-context learning. Although the idea itself is very simple, the experimental results obtained are also very impressive.

**Weaknesses:**

What everyone probably cares about is whether the generated demonstration is correct. The authors analyze this point in Sec. 4 and Appendix B, but the number of predictions analyzed is very small (20 correct and 20 incorrect), making it difficult to reach a statistically consistent conclusion.
In particular, we need a clear hypothesis as to why the correct answer rate increases even though the generated demonstration is incorrect, and a sufficient amount of evidence to support it.
Also, using only a single LLM is used in the experiment makes it difficult to discuss whether the quality of the results is based on the properties of the language model itself, or whether it is simply a property that only GPT 3.5 has in the current version.  In particular, recent papers on prompting often try to analyze universal results by comparing multiple language models.
As a science, it is important to show universal results in LLM to some extent.
The Chain-of-Thought paper also provides results for multiple LLMs.

**Questions:**

1. Can you analyze a sufficient number of results to prove your hypothesis regarding the following questions stated in your paper?
"Why incorrect few-shot demonstrations could lead to correct final predictions, while correct few-shot demonstrations could also lead to incorrect predictions?"

2.Can we analyze the generality of this result using multiple language models?

---

> ### Author Response · Authors · 2023-11-18
>
> We greatly appreciate the constructive feedback and insightful suggestions provided by the reviewer. Our responses to the raised concerns are as follows:
>
> **Q1**: The statistically consistent conclusion on the correctness of demonstrations
>
> **A1**: Sorry for the confusion. We acknowledge the reviewer's concern about the correctness of demonstrations. However, previous study shows that in-context learning does not rely on the input-label mapping in the demonstrations to perform the task (Min, Sewon, et al.). The correctness of individual demonstrations is not the sole determinant of the overall quality, especially in the context of language models' ability to generalize from these inputs.
>
> Thus, regarding the phenomenon that correct answer rate increases even though the generated demonstration is incorrect, a reasonable explanation is that strict correctness may not be the primary influencing factor in ICL's performance. Instead, language models might utilize other cues and patterns within demonstrations to arrive at correct conclusions.
>
> **Thus, the correctness of SEC's demonstration isn't a key factor of SEC's success.**
>
> Moreover, while our study involved 20 correct and 20 incorrect predictions, each case incorporated 5 demonstrations, resulting in an effective analysis of 200 demonstrations. This sample size, we believe, offers a reasonable overview of potential error types within demonstrations. Nevertheless, we recognize the importance of expanding this analysis and plan to include a larger dataset in future work to enhance the robustness of our conclusions.
>
> **Q2**: Generality of SEC using multiple language models
>
> **A2**: We concur with the reviewer on the importance of investigating SEC's applicability to multiple models. To this end, we intend to extend our experiments to include both open-source models and the newer GPT-4 architecture.
>
>
>
> [1] Min, Sewon, et al. "Rethinking the Role of Demonstrations: What Makes In-Context Learning Work?." Proceedings of the 2022 Conference on Empirical Methods in Natural Language Processing. 2022.

---

> > ### Author Response · Authors · 2023-11-21
> > **Update on the response about the experiments on multiple models**
> >
> > We have already carried out experiments of SEC and baselines on GPT4 and Llama2 34B. Please refer to the post with the title "Comparison between SEC and baselines on GPT4" and "Comparison between SEC and baselines on Llama2 34B".

---

> ### Comment · Reviewer_B8T2 · 2023-11-22
> **Update score**
>
> The author's reply and additional experiments seem reasonable to me.
> Therefore, I raise my score.

---

### Official Review · Reviewer_fqq3 · 2023-11-01

**Soundness:** 2 fair
**Presentation:** 3 good
**Contribution:** 3 good
**Rating:** 6
**Confidence:** 5

**Summary:**

The paper propose "self-contemplating (SEC)" prompting strategy a variation of the more common in-context learning (ICL) approach. The proposed approach consists of two steps: First, use the LLM to generate demonstrations based on the query sample; Second; use the generated demonstrations together with the query sample to create the final prompt that is fed back to the same LLM. The potential benefit lies in the fact that no additional reference training data is needed for curating the set of demonstrations. Experiments show that SEC performs comparably to the traditional ICL approach using the gpt-3.5-turbo model. However, SEC underperforms on other GPT models.

**Strengths:**

The paper is well written and is easy to understand. The paper introduces an interesting idea of only relying on the target LLM for generating demonstrations based on the target query sample. Doing so results in generating demonstrations that are probably better suited for the query sample. Also, the proposed approach helps remove the need for curating hand-crafted demonstrations which is a time consuming task.

**Weaknesses:**

1) The SEC method is only compared to the ICL approach where demonstrations are hand selected, i.e., not automatically selected. Several automated demonstration selection and curation approaches have been proposed in the last few years that should have been considered. SEC performs similar to hand-crafted ICL demonstrations. However, it is very likely that it might underperform once automated demonstration selection+curation approaches are introduced for comparison. Please look at the following papers and include them in your analysis:
 a) https://aclanthology.org/2023.findings-acl.273.pdf
 b) https://arxiv.org/abs/2211.04486
 c) https://arxiv.org/abs/2310.10707
 d) https://arxiv.org/abs/2302.05698
 e) https://arxiv.org/abs/2104.08786

3) Relying on the LLM for generating demonstrations has the potential problem of propagating biases that exist in the model. The SEC method does not account for situations where model's bias can impact the final prediction and overall result.

4) Only closed-source GPT-based models from OpenAI were considered. The authors should investigate if SEC can be extended to open-source models.

5) The authors claim that quality of demonstrations generated by the SEC paradigm are better than hand-crafted demonstrations. A qualitative analysis should be done to verify this claim. However, it is surprising to see that SEC performance reduces as number of demonstrations is increased, thereby contradicting this claim.

**Questions:**

Please see the above comments.

---

> ### Author Response · Authors · 2023-11-18
>
> We are thankful for the valuable insights and constructive feedback provided by the reviewer. We address each concern below:
>
> Sorry for the confusion. First we want to clarify that SEC isn’t a variant of in-context learning. Inherently, SEC is a zero-shot method which only relies on test input.
>
> **Q1**: Comparison with Automated Demonstration Selection Approaches
>
> **A1**: We apologize for any confusion. We acknowledge the reviewer's point on comparing SEC with automated demonstration selection and curation approaches. However, it's crucial to emphasize that **SEC inherently functions as a zero-shot method**, relying solely on test input, which sets it apart from ICL. Comparing SEC directly with automated demonstration selection approaches may not be entirely fair, **as the latter often requires a substantial pool of training examples for selection**. Our intention in comparing SEC with ICL was primarily illustrative, highlighting SEC's superiority over the zero-shot baseline.
>
> SEC substantially outperforms zero-shot baseline and reaches comparable performance with ICL methods which proof it’s efficacy. To provide a more comprehensive comparison, we are currently adding experiments with Zero-shot CoT. Preliminary results indicate that on benchmarks like GSM8k and ARC, Zero-shot CoT's performance (73.39 on GSM8k and 84.04 on ARC) is outperformed by CoT-SEC (77.0 on GSM8k and 86.9 on ARC).
>
> For the five paper you kindly provided, (a) focuses on exemplar selection from the view of explanations, (b) focuses on active example selection, (c) focuses on applying ICL to offensive content paraphrasing, (d) focuses on CEIL, an ICL example selection method, and (e) focuses addressing the order sensitivity in ICL. These approaches, indeed, improve upon vanilla ICL using training datasets. However, **our approach with SEC offers a novel perspective by achieving comparable ICL performance without relying on training data**. This difference in approach underscores SEC's flexibility and ease of use, presenting an alternative direction in the realm of language model prompting.
>
> **Q2**: Bias of SEC
>
> **A2**: We recognize the importance of addressing potential biases in LLM-generated demonstrations. While SEC demonstrates comparable performance to ICL in common benchmarks, we understand the need for further exploration, especially in tasks rare in training data. We plan to add such tasks in our revised version.
>
> Additionally, it's crucial to note that human-crafted prompts are not immune to biases. As highlighted by Ma et al. (2023), human biases can inadvertently influence the crafting of prompts. Hence, both human-crafted and LLM-generated content require careful consideration regarding bias.
>
> **Q3**: Applicability to Open-Source Models
>
> Our initial study focused on closed-source GPT-based models due to their prevalent use and performance. However, we agree with the reviewer that investigating SEC's applicability to open-source models is essential. We plan to conduct additional experiments with open-source models to validate SEC's effectiveness across a wider range of language model architectures.
>
> **Q4**: Quality of Demonstrations and Performance with Increased Demonstrations
>
> A4: Sorry for the confusion. We acknowledge that SEC's performance may not always surpass that of human-crafted prompts. The example in Figure 20 of our paper illustrates this point.
>
> Regarding the observed decrease in performance with an increased number of demonstrations, **we hypothesize the existence of an optimal number of demonstrations for SEC**, akin to what Reynolds and McDonell (2021) observed for human-crafted prompts. **The tailored nature of SEC's demonstrations might necessitate a smaller number of examples compared to human-crafted ones for optimal performance.** Therefore, **the decline in performance after a certain number of shots should not be misconstrued as an indication of poor demonstration quality**.
>
> Also, **we only observe this decrease in performance with an increased number of demonstrations on HumanEval dataset** while on other dataset (e.g. GSM8k), there doesn't exist such a decrease.
>
> Besides, we do include a **qualitative analysis on the correctness demonstrations in Section 4 and Appendix B** of our paper.
>
> [1] Ma, Huan, et al. "Fairness-guided Few-shot Prompting for Large Language Models." arXiv preprint arXiv:2303.13217 (2023).
>
> [2] Reynolds, Laria, and Kyle McDonell. "Prompt programming for large language models: Beyond the few-shot paradigm." Extended Abstracts of the 2021 CHI Conference on Human Factors in Computing Systems. 2021.

---

> > ### Author Response · Authors · 2023-11-21
> > **Update on the response about the experiments on multiple models and Zero-shot baselines**
> >
> > We have already carried out experiments of SEC and baselines on GPT4 and Llama2 34B. Please refer to the post with the title "Comparison between SEC and baselines on GPT4" and "Comparison between SEC and baselines on Llama2 34B".
> >
> > Also, in all our main experiments, we included Zero-shot CoT as one of our baselines to better evaluate the superior of SEC over zero-shot methods. SEC consistently outperforms Zero-shot CoT on all tasks we examined. Please refer to the revised version of our paper and the results of GPT4 and Llama2 34B.

---

> > > ### Comment · Reviewer_fqq3 · 2023-12-02
> > > **Updated Score**
> > >
> > > Thank you for your response and clarifications, I have updated my scores.

---

### Official Review · Reviewer_yTZX · 2023-11-07

**Soundness:** 3 good
**Presentation:** 3 good
**Contribution:** 2 fair
**Rating:** 6
**Confidence:** 3

**Summary:**

This paper proposes the self-contemplation prompting strategy (SEC) to let the LLM itself propose the demonstrations and then use the demonstrations to do downstream in-context learning. Experiments on different benchmarks show that the LLM itself can generate meaningful demonstrations to help improve performance.

**Strengths:**

- The paper is quite clear and easy to follow.

 - The method is easy to understand.

 - The evaluation is well-designed.

**Weaknesses:**

- The main concern is the significance of the paper. The proposed prompting method can be treated as a two-step chain-of-though prompting by letting the LLM (1) first think about the possible demonstration and then (2) use the demonstration to do in-context learning. Form this point of view, the prompting framework is one specific usage of CoT, which makes the contribution limited.

**Questions:**

- I am curious about the limitations of the proposed method. Using LLM for self-improvement may suffer from performance degradation – Once the LLM generates some wrong correction, the overall performance may drop significantly. Have you noticed any domains experiencing performance declines when using LLMs to generate their prompts?

---

> ### Author Response · Authors · 2023-11-12
>
> We would like to thank the reviewer for the encouraging comments and address the concerns below.
>
> **Q1:** On the Significance of SEC as Another Form of CoT
>
> **A1:** Your observation that SEC may seem akin to an extension of the chain-of-thought (CoT) prompting is insightful. However, SEC distinguishes itself by enabling the model to perform analogical reasoning, which is a leap beyond the capabilities of traditional CoT methods. SEC is fundamentally a zero-shot prompting pipeline. To provide a more comprehensive comparison, we are currently adding experiments with Zero-shot CoT. Preliminary results indicate that on benchmarks like GSM8k and ARC, Zero-shot CoT's performance (73.39 on GSM8k and 84.04 on ARC) is outperformed by CoT-SEC (77.0 on GSM8k and 86.9 on ARC).
>
> These findings reinforce our belief that SEC's performance cannot be easily replicated by existing CoT methods in a Zero-shot scenario. SEC's novelty, therefore, lies in its unique ability to autonomously generate meaningful demonstrations, which sets it apart from existing CoT approaches.
>
> **Q2:** The Limitations of SEC
>
> **A2:** We apologize for any confusion regarding the limitations of SEC. We recognize the potential for performance degradation with incorrect self-generated demonstrations. Our analysis indicates that large language models, particularly those with capacities at or above GPT-3.5, tend to be robust against minor errors in self-generated prompts for common tasks, maintaining performance comparable to that with human-crafted prompts.
>
> Our experiments across various reasoning tasks show that SEC generally achieves promising results. However, we recognize that in rare or abstract reasoning tasks, which have limited representation in training data, the model might encounter challenges. We plan to conduct further experiments in these domains to comprehensively understand SEC's limitations. Notably, in the domain of MATH, where LLMs typically struggle, CoT-SEC has outperformed CoT-ICL, reaffirming our observation about the robustness of larger models against minor prompt errors.
>
> In Section 4, we have evaluated SEC’s performance on smaller models and observed that its effectiveness diminishes with smaller model capacities. We are committed to extending our research with additional experiments using open-source models to further explore SEC’s applicability and limitations across different model scales.

---

> ### Author Response · Authors · 2023-11-21
> **Updates to the previous response**
>
> There are some updates to the response.
>
> **A1:** We have added zero-shot CoT as a part of our baseline methods to all our main experiments on GPT3.5, GPT4 and Llama2 34B. SEC consistently outperforms zero-shot CoT.
>
> The distinctiveness of SEC lies in its ability to autonomously generate meaningful demonstrations, differentiating it significantly from traditional CoT methodologies.
>
> **A2:** On the experiments of 200 3-digit base-5 addition problems, we do observe slight performance degradation of SEC compared to ICL. Nevertheless, SEC still outperforms it’s Zero-shot baselines in corresponding scenarios (Vanilla SEC to Zero-shot and CoT-SEC to Zero-shot CoT).
>
> Contrary to what might be assumed, the incorrect demonstrations are not the primary cause of this performance drop, as supported by the research of (Min, Sewon, et al.). We hypothesize that the performance degradation may stem from the model's unfamiliarity with the task, leading it to generate question-answer pairs that deviate from the expected distribution. For instance, the model might produce answers containing digits like 6, 7, 8, 9, which are outside the scope of base-5 calculations.
>
> #### References
> #### [1] Min, Sewon, et al. "Rethinking the Role of Demonstrations: What Makes In-Context Learning Work?." Proceedings of the 2022 Conference on Empirical Methods in Natural Language Processing. 2022.

---

> > ### Comment · Reviewer_yTZX · 2023-11-22
> > **Thank you for the rebuttal**
> >
> > I appreciate the author's detailed rebuttal, and my score remains unchanged.

---

### Author Response · Authors · 2023-11-18
**Comparison between SEC and baselines on GPT4**

Comparison between SEC and ICL in both CoT and answer only scenarios in **GPT4**.
| |      MATH    |   GSM8K  | ARC    | MMLU   | C-Eval      | HumanEval |
|----------------------|------|-------|-----|------|--------|------------|
|                      |      |       |   |      |        |            |
| **Published Results**||
| *Vanilla ICL*        | -    | - | 96.3$^a$  | 86.4$^a$  | [66.4$^c$] | [67.0$^a$] |
| *CoT-ICL*            | 42.6$^b$   | 92.0$^a$ |- | 86.4$^b$  |  68.7$^c$        | -          |
|                      |      |       |           |             |     |      |        |            |
| **Our Results**      ||
| *Zero-shot*          | 26.4 | 68.3  |  88.5  | 82.0   | 64.8   | 67.0$^a$      |
| *Zero-shot CoT*      | 32.6 | 86.7  | 90.2   | 82.2   | 64.4   | -       |
| *Vanilla ICL*        | 31.2 | 91.5  |  94.4  |**86.6**| 67.7   | **83.5**      |
| *Vanilla SEC*        | 35.0 | 91.7  |   94.7 | 86.1   |**68.1**| 83.0  |
|                      |(+3.8)|(+0.2) | (+0.3) | (-0.5) | (+0.4) | (-0.5)    |
| *CoT-ICL* |**42.3**|92.0$^a$| 95.1   | 86.0   | 67.0   | -          |
| *CoT-SEC*            | 41.9|**92.1**|**96.2**| 86.5   | 67.8   | -          |
|                      |(-0.4)|(+0.1) |(+1.1)  | (+0.5) | (+0.8) | -          |

SEC reaches comparable results to ICL.

Some of our results differ from the published results, which might be due to that the experiments was conducted on different model checkpoints. Any result encompassed within brackets signifies data derived from zero-shot prompting. The superscripts are used to indicate results that have been cited from previous studies: $^a$[1], $^b$[2], $^c$[3]. The results in the table will later be added into the revised version of our paper.

#### References
#### [1] OpenAI. Gpt-4 technical report, 2023.

#### [2] Yao Fu, Litu Ou, Mingyu Chen, Yuhao Wan, Hao Peng, and Tushar Khot. Chain-of-thought hub: A continuous effort to measure large language models’ reasoning performance. arXiv preprint arXiv:2305.17306, 2023.

#### [3] Yuzhen Huang, Yuzhuo Bai, Zhihao Zhu, Junlei Zhang, Jinghan Zhang, Tangjun Su, Junteng Liu, Chuancheng Lv, Yikai Zhang, Jiayi Lei, et al. C-eval: A multi-level multi-discipline chinese evaluation suite for foundation models. arXiv preprint arXiv:2305.08322, 2023.

---

### Author Response · Authors · 2023-11-18
**Limitations of SEC on underrepresented tasks**

Considering that SEC employs demonstrations that generated from LLMs, it may experience performance degradation in scenarios where the model is not strong enough, or the test data is not sufficiently represented in the training set. To investigate this issue, we design a  novel test set containing 200 3-digit base-5 addition problems which  appears rarely in everyday language and on web pages. We test SEC and baseline methods on this dataset. The results, as summarized in table, indicate that SEC tends to exhibit a slight decline in performance on these tasks compared to ICL methods.

| *Prompting Strategies*  |*Zero-shot* |*Zero-shot CoT*|*Vanilla ICL* | *CoT-ICL*  |*Vanilla SEC* | *CoT-SEC*    |
|:----:|:----:|:----:|:----:|:----:|:----:|:----:|
|                      |      |       |   |      |        |            |
|           Accuracy   |    26.5  |19.0|**28.0**| 27.0 |27.0| 24.0|

The results in the table will later be added into the revised version of our paper.

---

### Author Response · Authors · 2023-11-19
**Comparison between SEC and baselines on Llama2 34B**

####  Comparison between SEC and baselines in both CoT and answer only scenarios on **Llama2 34B**.

| Prompting Strategies | MATH | GSM8K | ARC | MMLU | C-Eval | Human Eval |
|:----:|:----:|:----:|:----:|:----:|:----:|:----:|
|                      |      |       |   |      |        |            |
| **Published Results**||
| *Vanilla ICL*        | 6.2$^a$   | 42.2$^a$| 54.5$^a$  | 62.6$^a$  | - | [22.6$^a$] |
|                      |      |       |           |             |     |      |        |            |
| **Our Results**      ||
| *Zero-shot*          | 3.5 | 29.7 |  54.4  | 56.1   | 36.5   | 19.5      |
| *Zero-shot CoT*      | 3.9 | 34.5  | 58.6   | 56.5   | 36.3   | -       |
| *Vanilla ICL*        | 6.4 | 42.0  |  67.2  |**62.3**| 38.5   | **22.5**      |
| *Vanilla SEC*        | 5.8 | 41.2  |   65.9 | 61.1   |39.0| 21.4  |
|                      |(-0.6)|(-0.8) | (-1.3) | (-1.2) | (+0.5) | (-1.1)    |
| *CoT-ICL*            | 7.4 |44.5| **68.7**   | 61.8   | 38.9   | -          |
| *CoT-SEC*            | **7.5**|**45.6**|67.5| 62.0   | **40.1**   | -          |
|                      |(+0.1)|(+1.1) |(-1.2)  | (+0.2) | (+1.2) | -          |

#### SEC reaches comparable results to ICL.

#### Some of our results differ from the published results, which might be due to that the experiments was conducted on different model checkpoints.  Any result encompassed within brackets signifies data derived from zero-shot prompting. The superscripts are used to indicate results that have been cited from previous studies: $^a$[1]. The results in the table will later be added into the appendix of our revised paper.

#### References
#### [1] Touvron, Hugo, et al. "Llama 2: Open foundation and fine-tuned chat models." arXiv preprint arXiv:2307.09288 (2023).

---

### Author Response · Authors · 2023-11-21
**Clarifications and Summarization of Contribution**

Upon reviewing our initial statements, we finds some misleading statements in the introduction, which we now clarify as follows:

* SEC is not a strengthen variation of ICL. Instead, SEC is a strong **Zero-shot** methods that performs comparably to and sometimes surpasses ICL.

* We emphasize that SEC is a zero-shot method, meaning it does not rely on in-domain adaptation. This characteristic is crucial to understanding its capabilities. **The fact that SEC achieves comparable results to ICL, without any in-domain adaptation, is a testament to its effectiveness and potential superiority in certain contexts.**

* According to Min, Sewon, et al., the correctness of demonstrations doesn’t have strong correlation to the performance of ICL. **Thus, the correctness of individual demonstrations is not the sole determinant of the overall quality**, especially in the context of language models' ability to generalize from these inputs. **The success of SEC is not because that the correctness of model-generated demonstrations is super high, which is also not true.**

We state the main contribution of our work:

* We proposed SEC, a strong zero-shot prompting methods, achieving comparable results to ICL on all common NLP tasks without **ANY** access to in-domain training data.

* SEC addressed the intensive work and difficulty when manually crafting demonstrations.

* SEC empowers even **inexperienced users** to solve problems with accuracy comparable to that achieved by ICL methods, enhancing the usability and accessibility of language model applications.

* **SEC’s performance totally relies on the model itself** (not conditioned on the selection and ordering of demonstrations), which can serve **a more comprehensive and stable pipeline** to evaluate the capability of LLMs.

* SEC’s comparable performance shows that with the generalization ability of LLMs to date, there is potential that **supervised training data may be dispensable in the future**.

* **SEC's demonstrations are tailored to each test cases**, which might lead to better performance considering the customization.

#### References
#### [1] Min, Sewon, et al. "Rethinking the Role of Demonstrations: What Makes In-Context Learning Work?." Proceedings of the 2022 Conference on Empirical Methods in Natural Language Processing. 2022.

---

### Author Response · Authors · 2023-11-21
**Summarization of Revision**

Dear Reviewers,

Thank you to all reviewers for their detailed and constructive comments. In response, we have made the following revisions to our paper:
* **Diverse Model Evaluation:** We have included results showcasing the effectiveness of SEC on both closed-source models like GPT4 and open-source models such as Llama2 34B, emphasizing SEC's wide-ranging applicability. Please refer to our post titled "Comparison between SEC and baselines on Llama2 34B" and "Comparison between SEC and baselines on GPT4".
*  **Inclusion of Zero-shot CoT as Baseline:** To facilitate a more robust comparison, we have incorporated Zero-shot Chain of Thought (CoT) as a baseline method. This enhancement allows for a clearer evaluation of SEC's performance relative to other Zero-shot methods.
* **Analysis of SEC and Auto-CoT:** We added the comparision of performance of SEC and Auto-CoT as a part of our ablation study.
* **SEC’s Limitations:** We conducted evaluations of SEC on less common tasks, specifically 3-digit base-5 addition problems, to highlight the method's limitations.
* **Clarifications and Refinements:** We have revised some previous statements for greater precision and rigor. For detailed information, please refer to our post titled "Clarifications and Summarization of Contribution."
* **Additional Minor Revisions:** For other minor changes, please refer to our responses to the reviewers' comments.

As the discussion phase nears its conclusion, we extend our gratitude for your invaluable insights.  We would greatly appreciate if you kindly let us know whether we have addressed all your concerns and your feedback on these updates. We're committed to addressing any remaining concerns and are ready to respond to any further questions you might have.


ICLR 2024 Conference Submission1674 Authors

---

### Meta-Review · Area_Chair_QD56 · 2023-12-07

**Metareview:**

The paper presents a useful approach, and one that opens up some interesting research questions.

There are some weaknesses that are critical for the authors to address (and they already started doing so in the discussion):
1. Experiments on more LLMs. Experiments on GPT4 only are definitely insufficient. Please expand your results beyond OAI models, especially to public research models (beyond APIs).
2. There has to stronger account of how this process may amplify bias or other adverse model tendencies.
3. Please move the discussion of generated demonstration correctness to the main paper, and expand it as much as possible. It’s an important and interesting aspect. What is the correlation between demonstration correctness and performance?

**Justification For Why Not Higher Score:**

Poster fits the level of contribution.

**Justification For Why Not Lower Score:**

Interesting and useful approach.

---

### Decision · Program_Chairs · 2024-01-16

Accept (poster)